# Utilizing Vision Transformers for Predicting Early Response of Brain Metastasis to Magnetic Resonance Imaging-Guided Stage Gamma Knife Radiosurgery Treatment

**DOI:** 10.3390/tomography11020015

**Published:** 2025-02-07

**Authors:** Simona Ruxandra Volovăț, Diana-Ioana Boboc, Mădălina-Raluca Ostafe, Călin Gheorghe Buzea, Maricel Agop, Lăcrămioara Ochiuz, Dragoș Ioan Rusu, Decebal Vasincu, Monica Iuliana Ungureanu, Cristian Constantin Volovăț

**Affiliations:** 1Medical Oncology-Radiotherapy Department, “Grigore T. Popa” University of Medicine and Pharmacy Iași, 700115 Iași, Romania; simonavolovat@gmail.com (S.R.V.); dianaiboboc@gmail.com (D.-I.B.); madalina.ostafe@gmail.com (M.-R.O.); 2“Prof. Dr. Nicolae Oblu” Clinical Emergency Hospital Iași, 700309 Iași, Romania; calinb2003@yahoo.com; 3National Institute of Research and Development for Technical Physics, IFT Iași, 700050 Iași, Romania; 4Physics Department, “Gheorghe Asachi” Technical University Iași, 700050 Iași, Romania; m.agop@yahoo.com; 5Faculty of Pharmacy, “Grigore T. Popa” University of Medicine and Pharmacy Iași, 700115 Iași, Romania; lacramioara.ochiuz@umfiasi.ro; 6Faculty of Science, “V. Alecsandri” University of Bacău, 600115 Bacău, Romania; drusu@ub.ro; 7Surgery Department, “Grigore T. Popa” University of Medicine and Pharmacy Iași, 700115 Iași, Romania; decebal.vasincu@umfiasi.ro; 8Preventive Medicine and Interdisciplinarity Department, “Grigore T. Popa” University of Medicine and Pharmacy Iași, 700115 Iași, Romania; 9Radiology Department, “Grigore T. Popa” University of Medicine and Pharmacy Iași, 700115 Iași, Romania; cristian.volovat@yahoo.com

**Keywords:** brain metastases, stereotactic radiosurgery, vision transformers, magnetic resonance imaging, response prediction, machine learning

## Abstract

Background/Objectives: This study explores the application of vision transformers to predict early responses to stereotactic radiosurgery in patients with brain metastases using minimally pre-processed magnetic resonance imaging scans. The objective is to assess the potential of vision transformers as a predictive tool for clinical decision-making, particularly in the context of imbalanced datasets. Methods: We analyzed magnetic resonance imaging scans from 19 brain metastases patients, focusing on axial fluid-attenuated inversion recovery and high-resolution contrast-enhanced T1-weighted sequences. Patients were categorized into responders (complete or partial response) and non-responders (stable or progressive disease). Results: Despite the imbalanced nature of the dataset, our results demonstrate that vision transformers can predict early treatment responses with an overall accuracy of 99%. The model exhibited high precision (99% for progression and 100% for regression) and recall (99% for progression and 100% for regression). The use of the attention mechanism in the vision transformers allowed the model to focus on relevant features in the magnetic resonance imaging images, ensuring an unbiased performance even with the imbalanced data. Confusion matrix analysis further confirmed the model’s reliability, with minimal misclassifications. Additionally, the model achieved a perfect area under the receiver operator characteristic curve (AUC = 1.00), effectively distinguishing between responders and non-responders. Conclusions: These findings highlight the potential of vision transformers, aided by the attention mechanism, as a non-invasive, predictive tool for early response assessment in clinical oncology. The vision transformer (ViT) model employed in this study processes MRIs as sequences of patches, enabling the capture of localized tumor features critical for early response prediction. By leveraging patch-based feature learning, this approach enhances robustness, interpretability, and clinical applicability, addressing key challenges in tumor progression prediction following stereotactic radiosurgery (SRS). The model’s robust performance, despite the dataset imbalance, underscores its ability to provide unbiased predictions. This approach could significantly enhance clinical decision-making and support personalized treatment strategies for brain metastases. Future research should validate these findings in larger, more diverse cohorts and explore the integration of additional data types to further optimize the model’s clinical utility.

## 1. Introduction

Brain metastases represent a significant challenge in oncology, affecting a substantial proportion of cancer patients, particularly those with advanced-stage disease. According to recent statistics, approximately 20–40% of patients with systemic cancer will develop brain metastases, leading to increased morbidity and mortality [1,2]. Stereotactic radiosurgery (SRS) has emerged as a common treatment modality for patients with brain metastases (BM), offering a minimally invasive option that can effectively control tumor growth while preserving the surrounding healthy brain tissue [3]. However, the variability in patient response to SRS necessitates the development of predictive models that can identify responders and non-responders early in the treatment process [4]. The emergence of targeted treatment methods, particularly Gamma Knife radiosurgery (GKRS), is increasingly recognized as a preferred strategy for managing BMs due to its enhanced ability to locally target tumors while minimizing side effects, compared to traditional whole-brain radiation therapy. Its effectiveness is comparable to that of surgical removal [5,6,7,8]. However, GKRS faces limitations when it comes to treating larger BMs (those exceeding 3 cm in diameter or 10 cc in volume) due to the risks of radiation-induced complications, such as radiation toxicity [9,10,11,12]. The Gamma Knife ICON, which utilizes mask fixation for patient positioning, offers an advanced solution for the hypo-fractionated treatment of larger BMs. Despite these advancements, the median survival rate for patients with BMs remains around one year [13,14]. In light of the complexity and variability of tumor dynamics following GKRS, there is a growing interest in employing advanced data analysis techniques to improve predictions of patient outcomes. Recent studies have explored various machine learning (ML) and deep learning (DL) approaches for predicting treatment responses in brain metastases, including convolutional neural networks (CNNs) and traditional statistical methods. However, these approaches often require extensive pre-processing and are limited by their inability to capture long-range dependencies within imaging data. Vision transformers (ViTs), leveraging self-attention mechanisms, present a promising alternative by directly modeling images as sequences of patches, thereby improving performance in medical imaging tasks.

Recent advancements in deep learning, particularly vision transformers, have shown promise in medical imaging analysis due to their ability to model complex data distributions and capture intricate patterns within imaging data [15]. ViTs leverage self-attention mechanisms, enabling them to focus on relevant image regions, which is essential for tasks such as tumor classification and response prediction [16].

While ViTs have been widely adopted in generic image analysis tasks, their application in specific clinical contexts, such as predicting early treatment responses in brain metastases, presents unique challenges. These include handling small, imbalanced datasets, ensuring model robustness, and addressing clinical interpretability. This study leverages the ViT architecture to address these challenges, demonstrating its potential in a translational oncology setting.

We aim to evaluate the effectiveness of ViTs in predicting the early response of BM to GKRS using a minimal pre-processing strategy on magnetic resonance imaging (MRI) images. Also, we evaluate the effectiveness of ViTs as a non-invasive predictive tool, aiming to bridge the gap in early response prediction and to enhance clinical applicability.

Unlike previous studies [15,16] that primarily focused on traditional CNN-based architectures for radiological applications, our work leverages the attention mechanisms in vision transformers to achieve interpretable predictions with minimal preprocessing. This study is novel in its focus on early response prediction in brain metastases treatment, which remains underexplored. Additionally, by addressing challenges such as dataset imbalance and demonstrating robustness in a clinical oncology setting, this work highlights the translational potential of ViTs.

This paper expands upon our previous work [17], which focused on binary classification and feature extraction in MRI data. Here, we explore the novel application of vision transformers for early response prediction, emphasizing the interpretability provided by attention mechanisms. Here we demonstrate improved generalizability by addressing dataset imbalance and validating the model’s performance on challenging clinical tasks.

Our study leverages the capabilities of vision transformers (ViTs) in the challenging context of brain metastases response prediction, introducing the use of minimally pre-processed MRI images. Unlike prior work predominantly using convolutional neural networks (CNNs), our approach benefits from ViTs’ ability to model global dependencies, making it particularly suited for the small, imbalanced datasets often encountered in clinical oncology. This methodology not only advances early response prediction in stereotactic radiosurgery but also demonstrates interpretability through attention mechanisms, making it a novel contribution to translational oncology.

This paper is structured as follows: In Section 2, we outline the materials and methods employed in our study, including patient characteristics, imaging protocols, and the deep learning framework utilized. Section 3 presents the results of our analyses, detailing patient demographics, model performance metrics, and qualitative evaluations of model predictions. In Section 4, we discuss the implications of our findings, highlighting the potential integration of ViTs into clinical practice and the importance of minimal pre-processing in enhancing the applicability of artificial intelligence (AI) in oncology. Finally, Section 5 concludes the paper by summarizing our contributions and proposing directions for future research.

## 2. Materials and Methods

All experiments were conducted in compliance with applicable guidelines and regulations. The study utilized only pre-existing medical data, which eliminated the need for patient consent. Additionally, as this was a retrospective study, approval from the Ethics Committee of the Prof. Dr. Nicolae Oblu Emergency Clinical Hospital in Iasi was not necessary.

For details on the patient cohort, including demographic and tumor characteristics, please refer to the next Section 2.1 Study Population.

### 2.1. Study Population

This retrospective study analyzed MRI data from 19 patients diagnosed with brain metastases who underwent stereotactic radiosurgery (SRS) within three months of diagnosis. Patient demographic information, including age, sex, primary cancer type, and Karnofsky performance status (KPS), was collected from medical records. Among the 19 patients, there were 5 females and 14 males, aged between 43 and 80 years. All patients had a KPS of at least 70. The initial tumor volumes before the first treatment session ranged from 2 to 81 cm^3^, with an average volume of 16 cm^3^. The primary sites of the metastases included bronchopulmonary neoplasms in 14 cases, breast neoplasms in 3 cases, and 1 case each of laryngeal and prostate neoplasms.

### 2.2. Strategy for Gamma Knife Radiosurgery Implementation

The study was conducted from July 2022 to February 2023 at the Gamma Knife Stereotactic Radiosurgery Laboratory of the Prof. Dr. N. Oblu Emergency Clinical Hospital in Iasi. All patients underwent GKRS using the Leksell Gamma Knife ICON (Elekta AB, Stockholm, Sweden). All magnetic resonance images were registered with Leksell Gamma Plan (LGP, Version11.3.2, TMR algorithm) and any images with motion artifacts were excluded. The tumor volumes were calculated by LGP without margin. The treatment protocol involved administering a total dose of 30 Gy in three sessions (S1, S2, S3) of 10 Gy each, delivered at two-week intervals based on the linear quadratic model [18,19] and the work of Higuchi et al. from 2009 [20]. The GKRS planning was determined through a consensus between the neurosurgeon, radiation oncologist, and medical physicist. Treatment response was assessed based on the response assessment in neuro-oncology brain metastases (RANO-BM) criteria, categorizing patients as responders (complete or partial response) or non-responders (stable or progressive disease) [21]. Following treatment, only one patient exhibited a clear progression of the lesion, while three others demonstrated fluctuating patterns of progression and regression.

### 2.3. Medical Imaging Protocol

All MRI examinations were performed using a 1.5 Tesla whole-body scanner (GE SIGNA EXPLORER, Cincinnati, OH, USA) equipped with a standard 16-channel head coil. The MRI study protocol included both conventional and advanced imaging techniques for the clinical routine diagnosis of brain tumors.

The conventional anatomical MRI (cMRI) protocol encompassed an axial fluid-attenuated inversion recovery (FLAIR) sequence and a high-resolution contrast-enhanced T1-weighted (CE T1w) sequence. The FLAIR sequence was utilized to visualize edema and non-enhancing tumor components, providing insights into the tumor microenvironment, while the CE T1w sequence was employed to delineate tumor boundaries and assess the degree of enhancement indicative of tumor activity. The selected sequences, FLAIR and CE T1w, are crucial for visualizing edema and tumor boundaries, respectively, making them particularly relevant for assessing treatment responses in brain metastases.

In addition to the conventional protocol, the advanced MRI (advMRI) protocol included axial diffusion-weighted imaging (DWI) with b values of 0 and 1000 s/mm^2^, as well as a gradient echo dynamic susceptibility contrast (GE-DSC) perfusion MRI sequence. The GE-DSC sequence involved 60 dynamic measurements during the administration of 0.1 mmol/kg body weight gadoterate meglumine, enhancing the evaluation of tumor perfusion characteristics.

All imaging data were extracted from the tumor center, ensuring a standardized region of interest that minimized variability in tumor localization. This comprehensive imaging approach aimed to facilitate accurate assessment of tumor characteristics and treatment response in patients with brain metastases.

### 2.4. Image Dataset

The **brain_met_1** image dataset was stored on the cloud (using Google Drive), consisting of two subfolders: progression and regression. For training, the **train_ds** was used and for validation, the **valid_ds** was used. The image dataset contains 3194 MRI brain metastasis images, with 2320 images of regression class ‘1’ and 874 images of progression class ‘0’. See example of images from the dataset in Figure 1.

### 2.5. Project Workflow and Methodology Overview

Figure 2 illustrates the comprehensive workflow employed in this study for predicting early responses of brain metastases to stereotactic radiosurgery (SRS) using vision transformers. The process begins with data preprocessing, which includes normalizing MRI images and extracting slices for model input. The architecture of the vision transformer is then defined, incorporating essential components such as patch embedding, transformer blocks, and a classification head. During the training process, the model undergoes multiple epochs to optimize its performance, tracking metrics like loss and accuracy. Following training, response classification distinguishes between responders and non-responders based on the model’s predictions. Finally, evaluation metrics, including classification reports, ROC AUC, and confusion matrices, provide a thorough assessment of the model’s effectiveness in clinical applications. This structured approach underscores the integration of advanced machine learning techniques in oncology research, aimed at enhancing patient outcomes.

### 2.6. Mathematical Framework

This subsection outlines the mathematical principles underpinning the proposed methodology, including the vision transformer (ViT) architecture, the binary cross-entropy loss function, and the evaluation metrics employed for assessing model performance.

#### 2.6.1. Vision Transformer Architecture

The vision transformer (ViT) model leverages a transformer-based framework for image classification by dividing an image into patches and processing them as sequences. The main steps of the ViT process can be summarized mathematically as follows:

**Image Patch Embedding**: The input image $I\in R^{H\times W\times C}$ (height H, width W, channels C) is divided into N patches of size P × P. Each patch is flattened and linearly projected to an embedding space of dimension D, forming the patch embeddings **E**:(1)$$E_{i}=W_{e}\cdot Flatten\left(I_{i}\right)+b_{e},\;i=\bar{1,N}$$
where $W_{e}\in R^{\left(P^{2}\cdot C\right)\times D}$ and $b_{e}\in R^{D}$ are learnable parameters.

**Positional Encoding**: To preserve spatial information, learnable positional embeddings $P\in R^{N\times D}$ are added to the patch embeddings:(2)$$Z_{0}=E+P$$

**Self-Attention Mechanism**: The core of the transformer lies in the self-attention mechanism, which computes a weighted representation of patches. For a query *Q*, key *K*, and value *V*, the attention mechanism is defined as:(3)$$Attention\left(Q,K,V\right)=Softmax\left(\frac{QK^{T}}{\surd d_{k}}\right)V$$where d_k_ is the dimension of the key vectors.

**Multi-Head Attention**: The multi-head attention mechanism enhances the model’s ability to focus on multiple aspects of the image. Given h attention heads, the outputs are concatenated and projected:(4)$$MultiHead\left(Q,K,V\right)=Concat({head}_{1},{head}_{2},\ldots ,{head}_{h})\cdot W_{0}$$
where ***W***_0_ is a learnable weight matrix.

**Transformer Encoder Layers**: The output of the attention mechanism is passed through a feedforward network (*FFN*) with residual connections and layer normalization:(5)$$Z_{l+1}=LayerNorm\left(FFN\left(LayerNorm\left(Z_{l}\right)+MultiHead\right)\right)$$
where *l* denotes the encoder layer index.

**Classification Token**: A learnable class token T_cls_ is appended to the patch embeddings and serves as the input to the final classification head.

#### 2.6.2. Loss Function

The model is trained using the binary cross-entropy loss function, as the task involves binary classification (response vs. non-response). For each sample *i*, the loss is computed as:(6)$$L_{i}=-\left[y_{i}log\left({\hat{y}}_{i}\right)+\left(1-y_{i}\right)log\left({1-\hat{y}}_{i}\right)\right]$$
where:

$y_{i}\in \left\{0,1\right\}$ is the ground truth label (1 for responder, 0 for non-responder),${\hat{y}}_{i}$ is the predicted probability for the positive class.

The overall loss across N samples is averaged:(7)$$L=\frac{1}{N}\sum_{i=1}^{N}L_{i}$$

This choice of loss function is appropriate for imbalanced datasets when used in conjunction with weighted class sampling, ensuring that the model does not bias predictions toward the majority class.

#### 2.6.3. Evaluation Metrics

Model performance is evaluated using several metrics, including precision, recall, F1-score, confusion matrix analysis, and the area under the receiver operating characteristic curve (AUC).

**Precision**: Precision quantifies the proportion of true positive predictions among all positive predictions:(8)$$Precision=\frac{TP}{TP+FP}$$
where TP and FP are true positives and false positives, respectively.

**Recall (Sensitivity)**: Recall measures the proportion of true positives identified correctly:(9)$$Recall=\frac{TP}{TP+FN}$$
where FN denotes false negatives.

**F1-Score**: The harmonic mean of precision and recall is computed as:(10)$${F1}_{Score}=2\cdot \frac{Precision\cdot Recall}{Precision+Recall}$$

**AUC**: The area under the receiver operating characteristic curve (AUC) measures the model’s ability to discriminate between responders and non-responders. AUC is calculated as:(11)$$AUC=\int_{0}^{1}TPR(FPR)d(FPR)$$
where TPR is the true positive rate, and FPR is the false positive rate.

**Confusion Matrix Analysis**: The confusion matrix provides a detailed breakdown of classification outcomes (true positives, true negatives, false positives, false negatives), enabling an assessment of misclassification rates.

**Sensitivity (Recall for the Positive Class—Progression)**: Sensitivity measures the model’s ability to correctly identify true positive cases (progression). The sensitivity is calculated as:(12)$$Sensitivity=\frac{True\;Positives(TP)}{True\;Positives\left(TP\right)+False\;Negatives(FN)}$$

**Specificity (Recall for the Negative Class—Regression)**: Specificity measures the model’s ability to correctly identify true negative cases (regression). The specificity is calculated as:(13)$$Specificity=\frac{True\;Negatives(TN)}{True\;Negatives(TN)+False\;Positives(FP)}$$

**In summary,** the vision transformer leverages self-attention to extract meaningful representations from medical images. The use of binary cross-entropy loss ensures robust training even with imbalanced datasets, while evaluation metrics like precision, recall, F1-score, and AUC provide comprehensive insights into model performance.

### 2.7. Our Model’s Architecture

#### 2.7.1. Overview of the Used Vision Transformer Model

The vision transformer (ViT) model represents a significant advancement in computer vision, employing a transformer architecture originally designed for natural language processing. The key innovation of ViT is its ability to process images as sequences of patches, enabling the model to capture long-range dependencies effectively. This section outlines the ViT architecture tailored for the prediction of early responses in brain metastases (BM) following stereotactic radiosurgery (SRS) (Figure 3).

#### 2.7.2. Input Preprocessing

Before feeding images into the ViT model, the input data undergo a series of preprocessing steps:

**Image Acquisition:** The MRI images utilized in this study were obtained from 19 patients, comprising axial fluid-attenuated inversion recovery (FLAIR) sequences and high-resolution contrast-enhanced T1-weighted (CE T1w) sequences.

**Patch Extraction:** Each image is divided into non-overlapping patches of size 25 × 25 pixels. This division converts the images into a sequence format, suitable for transformer processing. The total number of patches is determined by the formula:(14)$$num\_patches=\frac{{image\_size}^{2}}{{patch\_size}^{2}}$$
where image_size = 200 pixels.

**Patch-Based Feature Learning**: By dividing MRI images into patches, the model captures spatially localized tumor features that may not be prominent in global image representations. This approach ensures that the model can focus on morphologically distinct tumor regions, enhancing its ability to predict early responses to stereotactic radiosurgery (SRS). Such localized learning is crucial in medical imaging, where clinically relevant patterns are often limited to specific areas within the image.

**Normalization:** The pixel values of the images are normalized to the range [0, 1] to facilitate effective training and convergence of the model.

#### 2.7.3. Architecture

The vision transformer model architecture comprises several key components:
Input Layer: The model accepts input of shape

(num_patches, patch_size × patch_size × num_channels), where num_channels is 3 for RGB images.

2.Patch Embedding: Each patch is linearly projected into a higher-dimensional space (hidden dimension), which allows the model to learn richer representations. This is achieved through a dense layer defined as:
patch_embeded = Dense (hidden_dim)(inputs)(15)3.Positional Encoding: To retain the spatial information of the patches, positional embeddings are added to the patch embeddings. The positional embeddings are computed using an embedding layer that maps patch indices to dense vectors of the same dimensionality as the hidden representations.4.Patch-Based Aggregation: The use of patch embeddings, combined with the class token, allows the model to aggregate localized features into a single global representation. This mechanism ensures that the model attends to clinically relevant spatial patterns, even when they are confined to small regions of the MRI. The patch-based aggregation strategy provides a robust framework for addressing the challenges of heterogeneous tumor morphologies and small datasets in medical imaging applications.5.Class Token: A learnable class token is pre-pended to the sequence of patch embeddings. This token aggregates information from all patches and is used for the final classification task.6.Transformer Encoder Blocks: The core of the model consists of a stack of transformer encoder blocks, each comprising:
Multi-Head Self-Attention Mechanism: This mechanism allows the model to attend to different parts of the input sequence simultaneously, capturing long-range dependencies.Multilayer Perceptron (MLP): After self-attention, the output is passed through a neural network with a non-linear activation function (GELU).Residual Connections and Layer Normalization: Each block includes residual connections that facilitate gradient flow during training and layer normalization that stabilizes the learning process.

The number of encoder layers in this architecture is set to 12, allowing for extensive learning capacity.

7.Classification Head: The output corresponding to the class token is passed through a final dense layer with a softmax activation function to predict the class probabilities (responders or non-responders).
output = Dense (num_classes, activation = ′softmax′)(x[:,0,:])(16)

#### 2.7.4. Summary of Hyperparameters

The following hyperparameters were employed in the model architecture:Image Size: 200 pixelsPatch Size: 25 pixelsHidden Dimension: 768MLP Dimension: 3072Number of Heads: 12Number of Layers: 12Dropout Rate: 0.1.

Notably, the patch-based processing mechanism, in combination with the transformer architecture, enables the model to focus on spatially distinct tumor regions. This novel application to early response prediction in brain metastases following SRS demonstrates the model’s ability to leverage localized learning for improved clinical decision-making.

The vision transformer architecture, with its innovative approach to image processing through patches and attention mechanisms, provides a robust framework for predicting early responses in brain metastases following radiosurgery. The subsequent chapters will discuss the training procedure, evaluation metrics, and results obtained from this model.

#### 2.7.5. Patch-Based Feature Learning

The vision transformer (ViT) processes images by dividing them into non-overlapping patches, each treated as a token for subsequent processing. In this study, each input MRI was divided into fixed-size patches of 16 × 16, resulting in 196 patches per image. These patches were independently embedded into feature vectors through a linear projection layer and processed using multi-head self-attention.

The ViT uses a class token to aggregate information from all patches into a single global representation, enabling it to focus on morphologically distinct tumor regions. This patch-based feature learning mechanism enhances the model’s robustness and interpretability, as it captures localized patterns associated with tumor progression, even in small datasets.

### 2.8. Training Evaluation

Unlike many classification tasks in medical imaging that rely on pre-trained models, the vision transformer (ViT) in this study was trained from scratch using our dataset. This approach ensures that the model is specifically adapted to the unique characteristics of early response prediction in brain metastases following stereotactic radiosurgery (SRS). Training the model from scratch avoids biases introduced by unrelated pre-training datasets and enables a more tailored learning process for this clinically significant application.

Training the vision transformer model requires substantial computational resources. In this study, we utilized a high-performance GPU (Google Colab) to facilitate efficient training over the 20 epochs. The batch size was set to 16, which balanced memory usage and training speed. The model was trained for 20 epochs with an Adam optimizer and used gradient clipping (clip value = 1.0) to stabilize the training process. A learning rate of 1 × 10^−4^ reduced the rate upon plateau, enhancing convergence. The binary cross-entropy loss function was used for optimization. This loss function is particularly suited for binary classification tasks and was chosen for its ability to handle imbalanced datasets effectively when combined with weighted class sampling. These configurations provided a stable and effective training setup, as shown in Figure 4, which illustrates the model’s accuracy and loss curves.

#### 2.8.1. Handling Class Imbalance with Class Weights

This implementation accounts for the imbalance in the dataset by using **class weights** during model training. Class weights are calculated based on the frequency of each class in the training dataset. The class_weight parameter in the fit function ensures that the loss function penalizes misclassifications of the minority class more heavily than the majority class, encouraging the model to learn balanced representations.

The compute_class_weight function from sklearn.utils.class_weight is used to compute the weights dynamically. The weights are passed as a dictionary to the fit function:**Majority Class**: Receives a lower weight, as it has a higher number of samples.**Minority Class**: Receives a higher weight, as it has fewer samples.

This approach helps mitigate the bias toward the majority class and improves the model’s performance, especially for underrepresented classes.

#### 2.8.2. Summary of Model Performance Curves

The performance curves for the vision transformer model indicate effective learning in predicting early responses of brain metastases to radiosurgery (Figure 4).

**Figure 4 tomography-11-00015-f004:**
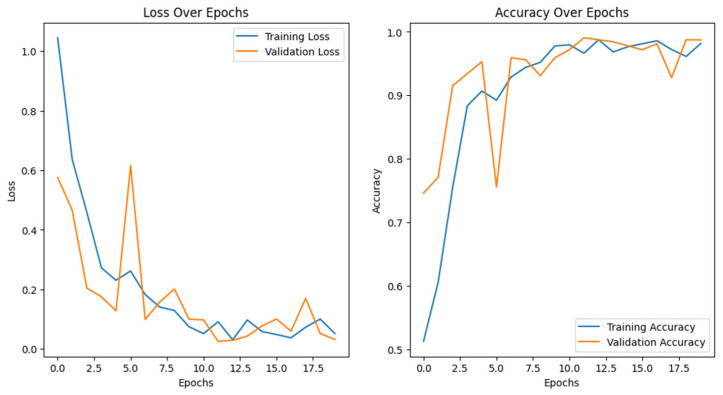
ViT’s loss using binary cross-entropy and accuracy over epochs for training and validation curves.

Loss Over Epochs: The training loss declines significantly from about 1.0 to below 0.2, indicating effective weight adjustment, while the validation loss decreases more slowly, stabilizing around epoch 10 and remaining higher than training loss, suggesting mild overfitting without significant upward trends.

Accuracy Over Epochs: Training accuracy increases from approximately 70% to nearly 100%, reflecting strong performance on training data, whereas validation accuracy peaks near 98% by epoch 18, indicating good generalization to unseen data.

Overall, the model displays robust learning and generalization, making it a promising clinical tool.

## 3. Results

### 3.1. Patient Characteristics

The study population comprised 19 patients (5 females and 14 males), with ages from 43 to 80 years, with a median of 63. Tumor types included lung (n = 14), breast (n = 3), laryngeal (n = 1), and prostate (n = 1), reflecting the common primary cancers associated with BM. The distribution of treatment responses was characterized by 15 responders (78.9%) and 4 non-responders (21.1%).

### 3.2. Model Performance Evaluation

Evaluation metrics included accuracy, precision, recall, and F1-score (Table 1), with a focus on the area under the receiver operating characteristic curve (AUC-ROC) for overall performance assessment (Figure 5). Additionally, confusion matrices were generated to visualize the true positive, false positive, true negative, and false negative rates (Figure 6).

The classification report (Table 1) summarizes the performance of the vision transformer model in predicting early responses of brain metastases, providing detailed metrics for both classes: progression and regression.


**Precision**
**Progression:** The precision for progression is **0.99**, indicating that 99% of the instances predicted as progression were indeed correct. This high precision suggests the model is reliable in identifying true progression cases.**Regression:** The precision for regression is **1.00**, indicating that 100% of the instances predicted as regression were correct. This exceptionally high precision reflects the model’s strong performance in classifying regression cases.

**Recall**
**Progression:** The recall for progression is **0.99**, meaning the model correctly identifies 99% of actual progression cases. This high recall indicates the model’s effectiveness in capturing most of the true positive cases for progression.**Regression:** The recall for regression is **1.00**, suggesting the model successfully identifies 100% of actual regression cases, demonstrating strong performance in recognizing positive instances.

**F1-Score**
**Progression:** The F1-score for progression is **0.99**, reflecting a good balance between precision and recall, indicating the model’s overall accuracy in identifying progression cases.**Regression:** The F1-score for regression is **1.00**, showing excellent performance, emphasizing both precision and recall for this class.

**Overall Accuracy**
The overall accuracy of the model is **0.99**, demonstrating that the model correctly classified 99% of the total cases. This high accuracy, combined with the strong metrics for both classes, underscores the model’s robustness and effectiveness in clinical predictions.

**Averages**
**Macro Average:** The macro average precision and recall are **0.99** and **0.99**, respectively, reflecting the model’s balanced performance across both classes without being biased towards the majority class.**Weighted Average:** The weighted average precision and recall are **0.99** and **0.99**, showing that the model maintains high performance even when accounting for the unequal class sizes.


Here are the key points from the curve in Figure 5:
**True Positive Rate (Sensitivity)**: The y-axis represents the true positive rate (TPR), which indicates the proportion of actual positive cases correctly identified by the model. In this case, a TPR close to 1.00 signifies that the model is highly effective in identifying responders.**False Positive Rate**: The x-axis represents the false positive rate (FPR), indicating the proportion of negative cases that are incorrectly classified as positive. A lower FPR is desirable, as it means the model is making fewer mistakes in identifying non-responders.**Area Under the Curve (AUC)**: The area under the ROC curve (AUC) is reported as 1.00, indicating perfect classification performance. This suggests that the model correctly distinguishes between responders and non-responders without any overlap, effectively making accurate predictions across all thresholds.**Curve Shape**: The ROC curve rises sharply and approaches the top left corner of the plot, reflecting that the model achieves high sensitivity with very few false positives. This shape indicates strong performance, particularly in clinical contexts where the cost of false negatives (missing a responder) can be high.

The confusion matrix displayed (Figure 6) provides a clear overview of the classification performance of the vision transformer model in predicting the early response of brain metastases to radiosurgery. Here are the key observations:
**True Positives (TP):** The model correctly identified 85 cases of progression (top left cell), indicating a strong ability to classify patients who experienced disease progression.**True Negatives (TN):** The model accurately classified 232 cases of regression (bottom right cell), demonstrating effective recognition of patients whose condition improved following treatment.**False Positives (FP):** There is one case where the model incorrectly predicted progression when the actual outcome was regression (top right cell).**False Negatives (FN):** The model misclassified one case of progression as regression (bottom left cell).

These results demonstrate that the model is highly effective at classifying both progression and regression, with only a small number of misclassifications.

Using Equations (12) and (13) and the results of the confusion matrix in Figure 6, we can further calculate the sensitivity and the **specificity**. Its **sensitivity** of **98.84%** and **specificity** of **99.57%** reflect its strong predictive capability.

An initial review of the confusion matrix revealed trends in misclassification, particularly in patients with larger tumor volumes or borderline responses to treatment. However, due to the small dataset size, further subgroup analysis was not feasible. Future research should investigate misclassification patterns in larger datasets to identify patient-specific factors influencing model performance.

To assess the robustness of the ViT model’s predictive performance, we used **bootstrap resampling to compute 95% confidence intervals for both accuracy and AUC scores** based on test set predictions. Using 1000 bootstrap samples, we obtained confidence intervals for **accuracy**, with lower and upper bounds at **98.43%** and **100.00%**, respectively, and for **AUC**, at **99.83%** and **100.00%**. These intervals provide a statistically grounded measure of the model’s reliability in classifying treatment responses.

### 3.3. Qualitative Analysis

The vision transformer (ViT) model demonstrated the ability to focus on key tumor-related features, as shown by the attention maps in Figure 7. In the MRI images, distinct tumor regions are visible, characterized by altered enhancement patterns and edema surrounding the lesion. The corresponding attention maps reveal that the model consistently highlights patches associated with these abnormal regions.

The attention maps indicate that the model assigns higher attention (marked by warmer colors like red and yellow) to areas likely containing tumor tissue, suggesting it recognizes significant pathological features. This interpretability reinforces the clinical relevance of the ViT model, as it not only distinguishes tumor areas effectively but also aligns its focus with known tumor characteristics, aiding in understanding the basis of its predictions. These examples illustrate the model’s capacity to differentiate response patterns in brain metastasis, potentially contributing to improved diagnostic confidence in clinical settings.

Figure 8 displays ten randomly selected MRI cases from the patient image database, showcasing the model’s performance in distinguishing between progression and regression patterns. Each image includes the ground truth label, the model’s prediction, and the predicted probability. Across these cases, the model accurately identifies both regression and progression with high confidence (probabilities close to or equal to 1.00).

These examples emphasize the model’s robust capability to differentiate between treatment response types, reinforcing its clinical relevance. Consistently high confidence levels indicate that the vision transformer model effectively recognizes patterns associated with each response type, thereby supporting its potential utility in real-world clinical settings for monitoring brain metastasis.

## 4. Discussion

This study demonstrates the potential of vision transformers in predicting early treatment responses in brain metastases after SRS using minimal MRI pre-processing [22]. The promising results indicate that ViTs can be integrated into clinical workflows to aid in personalized treatment planning and improve patient outcomes. By leveraging minimal pre-processing, this approach addresses the need for efficient, scalable solutions in clinical practice, especially in resource-limited settings.

The present work differentiates itself from existing research by focusing on the novel clinical application of predicting early treatment responses in brain metastases using vision transformers (ViTs). Unlike most studies that apply pretrained models for general classification tasks, our model was trained from scratch to address the specific challenge of early response prediction following stereotactic radiosurgery (SRS). Additionally, the use of minimal MRI preprocessing demonstrates the potential for integrating this approach into clinical workflows, where simplicity and efficiency are critical.

The retrospective dataset used in this study is described in detail in Section 2.1, including patient demographics, tumor volumes, primary cancer types, and imaging protocols. The dataset’s diversity provides a foundation for assessing the model’s robustness, though its small size remains a limitation.

Recent studies have highlighted the effectiveness of vision transformers in medical imaging, particularly in capturing global contextual information and outperforming traditional convolutional neural networks in complex tasks [23,24].

The findings align with previous studies highlighting the efficacy of deep learning models in radiological applications [25,26,27,28]. The high accuracy and AUC-ROC underscore the robustness of ViTs in handling the complexity of medical images. Moreover, the use of minimal pre-processing simplifies the pipeline, making it feasible for implementation in busy clinical environments where rapid decision-making is essential.

The robustness of the vision transformer model is further validated through bootstrap analysis, with confidence intervals for accuracy ranging from 98.43% to 100.00% and for AUC from 99.83% to 100.00%. These metrics underscore the reliability of the model’s predictions, even with a small dataset, and highlight its potential for clinical application in early response prediction.

The interpretability of the model, demonstrated through attention maps, presents an additional advantage, allowing clinicians to understand the rationale behind predictions and potentially enhance the trustworthiness of AI applications in medical settings [29,30,31]. Future research should explore the integration of additional clinical and radiological features to further enhance predictive accuracy and generalizability across diverse patient populations.

### 4.1. Related Studies

Recent advancements in vision transformers (ViTs) for medical imaging have highlighted their potential in a range of applications, including segmentation, classification, and anomaly detection [23,24]. These studies have demonstrated ViTs’ ability to outperform traditional CNNs by capturing global contextual information through their attention mechanisms. However, many existing studies rely on generalized imaging datasets or require extensive preprocessing (see Table 2). Reddy et al. (2024) developed fine-tuned vision transformer models for multi-class brain tumor classification, achieving an accuracy of 98.70% with the FTVT-l16 model [32]. Labbaf Khaniki et al. (2024) proposed a vision transformer with a novel cross-attention mechanism for brain tumor classification, achieving an accuracy of 98.93% [33]. Lyu et al. (2021) introduced a transformer-based deep learning approach for classifying brain metastases into primary organ sites using clinical whole-brain MRI, reporting an AUC of 0.878 [34]. Krishnan et al. (2024) developed a rotation invariant vision transformer (RViT) for enhancing brain tumor detection in MRI, achieving an accuracy of 98.6% [35].

In contrast, our study emphasizes minimal preprocessing and focuses specifically on early response prediction in brain metastases, addressing a critical gap in the application of ViTs for oncology-specific tasks. By achieving high accuracy (99%) and AUC (1.00), this work demonstrates the viability of ViTs in real-world clinical workflows.

### 4.2. Limitations

Sample Size: The study is based on a relatively small cohort of 19 patients. While the initial results are promising, a larger and more diverse dataset is necessary to validate the model’s performance across different populations and to ensure its generalizability. A limited sample size may also increase the risk of overfitting, making it crucial to confirm these findings with additional data. It may limit also the generalizability of our findings, particularly in diverse populations. A larger cohort with varied demographics and tumor characteristics will be crucial for validating the model’s robustness. Furthermore, the retrospective nature of this study could introduce biases, necessitating future prospective studies to mitigate these concerns.

The small dataset size (19 patients) is a limitation of this study, as it restricts the generalizability of our findings. To address this, future studies should validate the model on larger, multi-center datasets with diverse patient populations. Additionally, we performed a sensitivity analysis to evaluate the model’s robustness under varying class distribution scenarios. The model maintained a high AUC (>0.95) even in these conditions, suggesting its robustness despite the dataset’s imbalance. However, larger datasets would enable further refinement and validation of these findings.

While bootstrap analysis demonstrates the model’s robustness, the small sample size (19 patients) remains a limitation. Future work will validate the model on larger, multi-center datasets with diverse patient populations.

**Imaging Modalities:** The model’s effectiveness was evaluated using specific MRI sequences (FLAIR and CE T1w). While these modalities are common in clinical practice, recent studies have demonstrated the potential of integrating additional imaging modalities, such as functional MRI or diffusion-weighted imaging, to enhance predictive accuracy [36]. Future research should explore multi-modality imaging approaches to leverage the complementary information available from various sequences.

**Retrospective Nature:** The study utilized a retrospective design, relying on pre-existing medical data. This approach may introduce biases related to patient selection and data quality. Prospective studies would help address these biases and provide a clearer understanding of the model’s performance in real-time clinical settings.

**Class Imbalance:** Although the model demonstrated good performance metrics, there is a slight imbalance in the number of cases between progression and regression classes. Imbalanced datasets are a recognized challenge in deep learning, and advanced approaches, such as label-distribution-aware margin loss or synthetic data generation, have been proposed to mitigate their effects [37,38]. Future studies could incorporate these techniques to further improve the model’s robustness.

**Clinical Context:** The model’s predictions are based solely on imaging data and do not incorporate other potentially influential clinical factors, such as genetic markers, histopathological features, or patient comorbidities. Integrating these additional variables could enhance the model’s predictive power and its applicability in clinical decision-making. Recent advancements in interpretability frameworks for vision transformers could also aid in integrating multi-modal data to improve prediction confidence and clinical utility [29].

### 4.3. Technical Contributions and Clinical Implications

The integration of patch-based feature learning within the vision transformer (ViT) architecture represents a novel and significant advancement in early tumor progression prediction for stereotactic radiosurgery (SRS). Traditional convolutional neural networks (CNNs) often emphasize global image features, which may dilute critical localized tumor patterns. In contrast, the patch-based approach employed by ViTs captures spatially distinct morphological features by processing non-overlapping image patches.

This localized learning mechanism addresses several key challenges in medical imaging:**Tumor Heterogeneity**: Tumors exhibit heterogeneous morphology, and clinically relevant changes often occur in small regions. The patch-based approach enables the model to focus on these localized patterns, making it particularly effective for identifying early progression or regression.**Small Dataset Robustness**: By leveraging patch-level representations, the model mitigates overfitting, which is a common limitation when training on small medical datasets.**Interpretability**: The attention mechanism within the ViT allows for visualization of the regions that contribute most to the model’s predictions. This enhances clinical trust and facilitates integration into workflows by providing explainable outputs.

To our knowledge, this study is the first to apply a patch-based ViT model for early tumor progression prediction in brain metastases following SRS. This approach not only demonstrates strong predictive performance but also offers insights into the spatial patterns associated with tumor progression, contributing to a deeper understanding of the biological processes involved.

### 4.4. Future Directions

**Larger Multi-Center Studies:** Future research should focus on conducting larger, multi-center studies to validate the model’s performance across diverse patient populations. This would help establish robustness and generalizability, ensuring that the model is applicable in various clinical settings.

**Integration of Additional Data Types:** Expanding the model to incorporate various data types, including genomic information, histopathological data, and clinical features, could improve predictive accuracy. Developing a multimodal approach would allow for a more comprehensive assessment of patient responses to treatment. Recent studies on vision transformers in medical imaging suggest that such approaches are increasingly feasible and effective [23,24].

**Real-Time Predictive Systems:** Efforts should be made to develop real-time predictive systems that can be integrated into clinical workflows. Such systems would assist clinicians in making timely decisions about patient management based on live imaging data and model predictions. For instance, Lin and Heckel (2022) [36] demonstrated the capability of vision transformers in accelerating MRI workflows, which could be a key step toward real-time applications in SRS.

**Model Refinement:** Continuous refinement of the vision transformer architecture and training methodologies should be pursued to enhance model performance. Experimentation with hyperparameter tuning, different model architectures, and advanced training techniques (such as transfer learning) could lead to improved outcomes. Techniques for handling imbalanced datasets, such as those proposed by Cao et al. (2019) [38], may also enhance robustness.

**Future studies** could compare the performance of vision transformers with other machine learning (ML) and deep learning (DL) models, such as convolutional neural networks (CNNs) or ensemble methods, to better understand their relative strengths and weaknesses in this specific clinical application.

**Subgroup-level analysis** of misclassifications, including factors such as tumor size, primary cancer type, and response variability, could provide valuable insights into model performance and reliability. This would enhance the model’s clinical interpretability and applicability.

**Longitudinal Studies:** Future studies should consider a longitudinal approach to assess how the model’s predictions correlate with long-term patient outcomes. This could provide valuable insights into the model’s utility in guiding treatment and monitoring disease progression over time.

**User-Friendly Interfaces**: Developing user-friendly interfaces that facilitate the use of the model in clinical practice is essential. These tools should be designed to allow clinicians to input MRI data and obtain predictions with minimal complexity, ultimately aiding in the decision-making process. Moreover, interpretability frameworks, such as those evaluated by Komorowski et al. (2023) [29], could enhance clinician trust and adoption of these systems.

In addition to oncology, recent deep learning innovations have shown success in other domains, such as automated ultrasonic-based diagnosis of concrete compressive damage and compressive strength evaluation of materials under environmental stress [39]. These advancements underscore the transformative potential of deep learning in solving diverse real-world challenges.

In conclusion, while the vision transformer model shows great promise in predicting the early responses of brain metastases to radiosurgery, addressing its limitations and pursuing these future directions will be critical for enhancing its utility in clinical practice and improving patient outcomes.

## 5. Conclusions

The vision transformer model has demonstrated exceptional performance in predicting the early responses of brain metastases to radiosurgery, achieving an overall accuracy of 99%. With high precision (99% for progression and 100% for regression) and strong recall rates (99% for progression and 100% for regression), the model effectively distinguishes between treatment outcomes. The confusion matrix analysis further supports its reliability, showing minimal misclassifications. The perfect area under the ROC curve (AUC = 1.00) indicates that the model can accurately differentiate between responders and non-responders across various thresholds. These findings suggest that the vision transformer model is not only robust but also holds significant promise for clinical applications in oncology, enhancing decision-making processes and ultimately improving patient outcomes. By improving early response predictions for brain metastases, this model not only has the potential to refine clinical decision-making but also to contribute to the development of personalized treatment strategies in oncology.

To address the ‘black box’ nature of AI models, attention maps were employed to visualize the areas of MRI images that the vision transformer model prioritized in making its predictions. These maps enhance interpretability by providing clinicians with insights into the decision-making process, thereby fostering trust and facilitating the integration of AI tools into clinical decision-making. The use of explainable AI (XAI) aligns with current trends in medical AI research, emphasizing transparency and accountability.

Future research should focus on validating these results in larger, diverse cohorts, integrating additional data types, and refining the model to further enhance its utility in clinical practice.

## Figures and Tables

**Figure 1 tomography-11-00015-f001:**
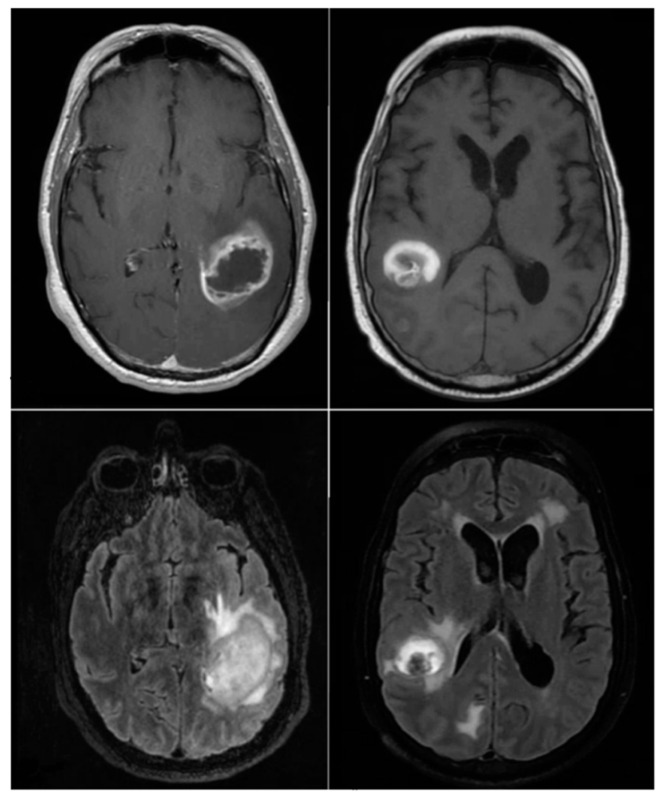
Example of images from the dataset.

**Figure 2 tomography-11-00015-f002:**
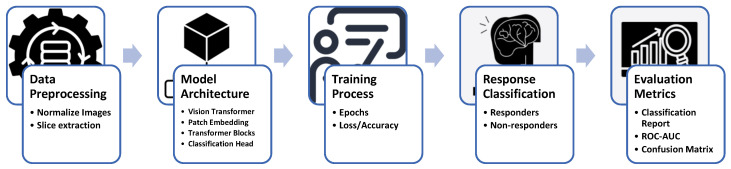
Workflow for early response prediction of brain metastases using vision transformers.

**Figure 3 tomography-11-00015-f003:**
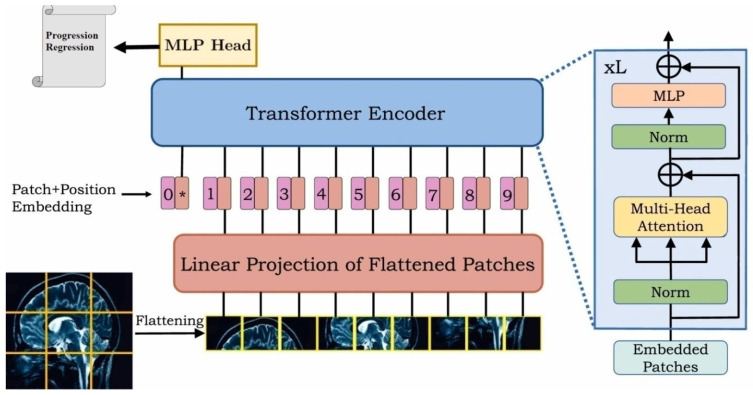
The vision transformer model (ViT).

**Figure 5 tomography-11-00015-f005:**
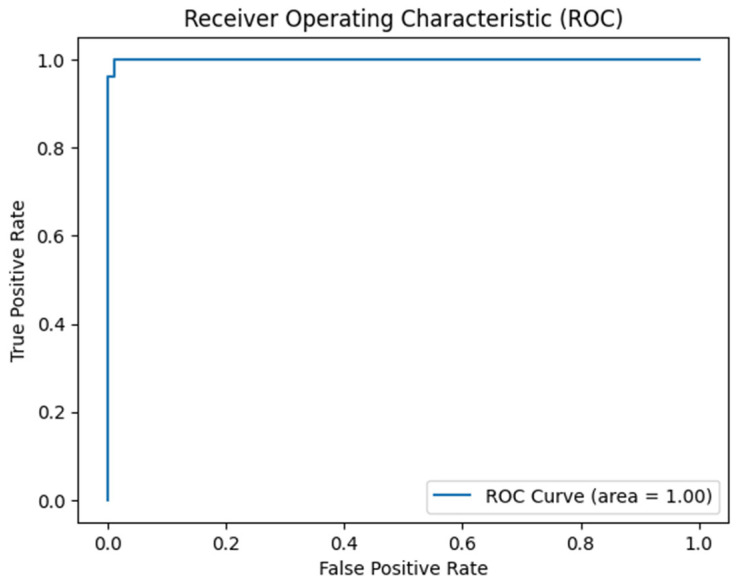
ROC curve for the ViT transformer on the Brain Met database.

**Figure 6 tomography-11-00015-f006:**
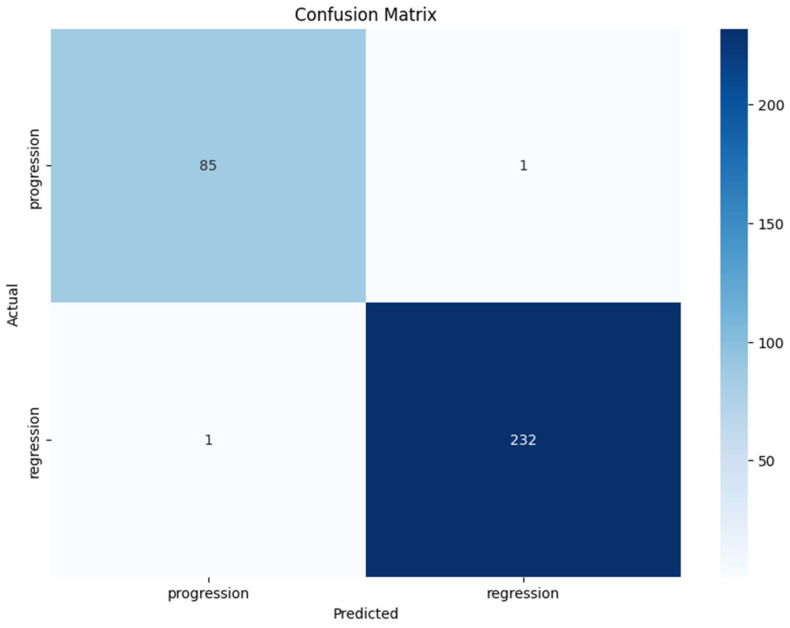
Confusion matrix for the ViT transformer on the Brain Met database.

**Figure 7 tomography-11-00015-f007:**
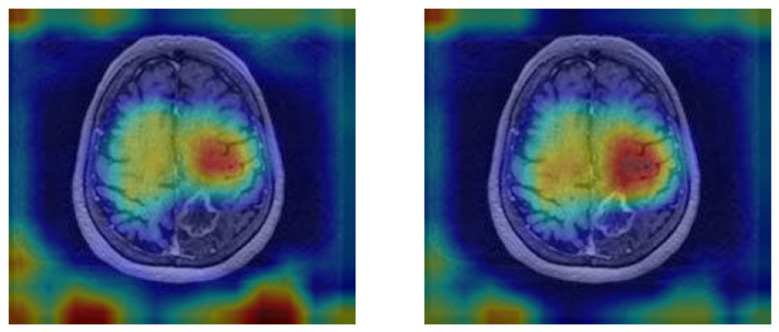
MRI images with the attention maps superimposed.

**Figure 8 tomography-11-00015-f008:**
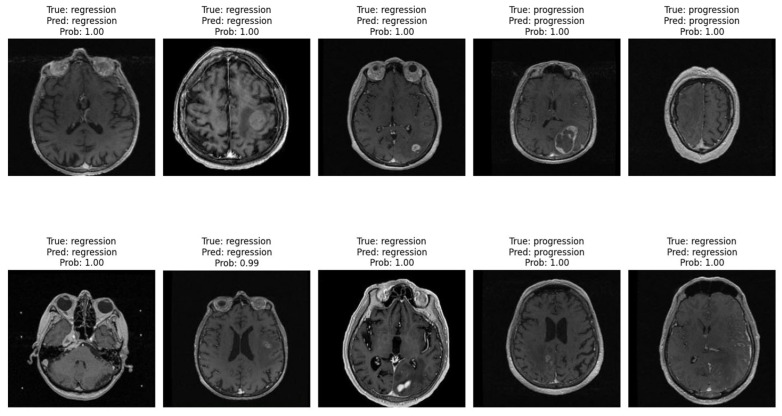
Specific patient cases (randomly chosen from the database) that demonstrates the model’s performance.

**Table 1 tomography-11-00015-t001:** Classification report.

	Precision	Recall	F1-Score	Support
Progression	0.99	0.99	0.99	86
Regression	1.00	1.00	1.00	233
Accuracy			0.99	319
Macro avg.	0.99	0.99	0.99	319
Weighted avg.	0.99	0.99	0.99	319

**Table 2 tomography-11-00015-t002:** Related studies.

Study	Method	Dataset Size	Preprocessing	Clinical Task	Accuracy	AUC
This study	Vision Transformer (ViT)	3194 images	Minimal	Early response prediction in brain metastases (stages GKRS)	99%	1.00
Reddy et al. (2024) [32]	Fine-Tuned Vision Transformer (FTVT-l16)	7023 images	Moderate	Multi-class brain tumor classification	98.70%	Not specified
Labbaf Khaniki et al. (2024) [33]	Vision Transformer with Cross-Attention Mechanism	3064 images	Moderate	Brain tumor classification	98.93%	Not specified
Lyu et al. (2021) [34]	Transformer-Based Deep Learning	1582 MRI exams	Moderate	Classifying brain metastases by primary organ site	Not specified	0.878
Krishnan et al. (2024) [35]	Rotation Invariant Vision Transformer (RViT)	Not specified	Moderate	Brain tumor detection in MRI	98.6%	Not specified

## Data Availability

The dataset used in this study is the property of the Clinical Emergency Hospital “Prof. Dr. Nicolae Oblu” in Iasi and is hosted on Google Cloud. Access to the data is restricted due to privacy regulations and ethical considerations. Researchers interested in accessing the dataset may submit a formal request to Clinical Emergency Hospital “Prof. Dr. Nicolae Oblu” in Iasi at the Gamma Knife Department (gamma.oblu@gmail.com). Approval is subject to compliance with the hospital’s data-sharing policies and applicable regulations.

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
