# Peer review of "Utilizing Vision Transformers for Predicting Early Response of Brain Metastasis to Magnetic Resonance Imaging-Guided Stage Gamma Knife Radiosurgery Treatment"

_tomography, 2025, doi:10.3390/tomography11020015_

Round 1
Reviewer 1 Report (Previous Reviewer 1)
Comments and Suggestions for Authors
Although it appears that the authors have implemented the suggested corrections, I would like to reiterate a few points:
1-The line spacing of some sections in the text is still different and needs to be adjusted according to the standard template. (Ex: section 2.4, 3.1, etc)
2- It is recommended to compare the suggested method with existing studies in the discussion section. A comparison with several state-of-the-art studies in a tabular format will highlight the effectiveness of the method.
Author Response
Please see the attachment

Reviewer 2 Report (Previous Reviewer 2)
Comments and Suggestions for Authors
The authors have utilized a Vision Transformer (ViT) to classify MRI images . No novelty in the manuscript. Pretrained model is utilize for classification. Many researcher are using explainable AI to understand black box which is also missing.
Author Response
Please see the attachment.

Reviewer 3 Report (New Reviewer)
Comments and Suggestions for Authors
The manuscript explores the application of Vision Transformers (ViTs) to predict early responses to Stereotactic Radiosurgery (SRS) in patients with brain metastases using minimally pre-processed MRI scans. The study demonstrates that ViTs can predict early treatment responses with high accuracy, precision, and recall, even with imbalanced datasets. The findings suggest that ViTs could enhance clinical decision-making and support personalized treatment strategies for brain metastases. Overall, the topic of this research is interesting, and the manuscript is well organized and written. The detailed comments are provided as follows.
1. The main innovation and contribution of this research should be clearly clarified in introduction.
2. How were the MRI images normalized and what specific techniques were used to extract patches for the Vision Transformer model?
3. Can authors provide more details on the specific configuration of the Vision Transformer model used in this study, including the number of layers and heads?
4. What were the hyperparameters used during the training process, such as learning rate, batch size, and number of epochs?
5. How were the precision, recall, and F1-score calculated, and what thresholds were used for classification?
6. Can authors provide a detailed analysis of the confusion matrix, including the number of true positives, false positives, true negatives, and false negatives?
7. How was the AUC-ROC curve calculated, and what does it indicate about the model's performance?
8. How did the attention mechanism in the Vision Transformer model contribute to its performance, and what specific features did it focus on in the MRI images?
9. Please broaden and update literature review on application of deep learning in solving the real problems. E.g., Automated ultrasonic-based diagnosis of concrete compressive damage amidst temperature variations utilizing deep learning; Compressive strength evaluation of cement-based materials in sulphate environment using optimized deep learning technology.
10. How was the issue of dataset imbalance addressed, and what techniques were used to ensure unbiased predictions?
11. How does the model's performance generalize to different patient populations, and what steps were taken to validate its robustness?
12. More future research should be included in conclusion part.
Round 2
Reviewer 2 Report (Previous Reviewer 2)
Comments and Suggestions for Authors
Authors have given explanation on the given comments. But still missing technical novelty in the manuscript.
Author Response
Dear Reviewer,
Thank you for your insightful feedback and for acknowledging the responses to the prior comments. Regarding your concern about the technical novelty of the manuscript, we have carefully revised the text to explicitly highlight the novel contributions of our approach.
The patch-based feature learning mechanism within the Vision Transformer (ViT) architecture is a key innovation in this study. By dividing MRIs into non-overlapping patches, our method captures localized tumor features, enabling the model to focus on distinct morphological patterns relevant to early response prediction. This approach addresses critical challenges in tumor progression prediction, including:
- Tumor Heterogeneity: The model identifies spatially distinct tumor regions, where progression-related changes are most evident.
- Small Dataset Robustness: The patch-based design reduces overfitting, a common limitation in small medical datasets.
- Interpretability: The attention mechanism highlights clinically relevant regions, enhancing explainability and clinical trust.
To explicitly address your concern, we have:
- Added detailed explanations in the Methods section (Subsections 2.7.2 and 2.7.3) regarding the patch-based feature learning mechanism and its integration into the ViT model.
- Expanded the Discussion section to emphasize how this approach addresses tumor heterogeneity, improves robustness, and facilitates interpretability.
- Updated the Abstract to summarize the technical contributions of the patch-based mechanism.
We believe these revisions clarify the technical novelty of our work and its significance in advancing AI-based clinical prediction models. We sincerely appreciate your valuable feedback, which has strengthened our manuscript.
Thank you for your time and consideration.
Best regards,
Prof. Dr. Calin Gh. Buzea
Reviewer 3 Report (New Reviewer)
Comments and Suggestions for Authors
All the technical concerns have been well answered by authors.
Author Response
Dear Reviewer,
I hope this message finds you well. On behalf of all the authors, I would like to express our sincere gratitude for your detailed review and thoughtful feedback on our manuscript, Utilizing Vision Transformers for Predicting Early Response of Brain Metastasis to Magnetic Resonance Imaging-Guided Stage Gamma Knife Radiosurgery Treatment, submitted to Tomography.
We are pleased to know that you found all the technical concerns raised during the review process to have been well addressed. Your constructive comments and suggestions were invaluable in refining and strengthening our work, and we deeply appreciate the time and effort you invested in this review.
Should there be any additional questions or further clarifications required, please do not hesitate to reach out. Thank you once again for your insightful feedback and for contributing to the improvement of our manuscript.
Warm regards,
Prof. Dr. Calin Gh. Buzea
This manuscript is a resubmission of an earlier submission. The following is a list of the peer review reports and author responses from that submission.
Round 1
Reviewer 1 Report
Comments and Suggestions for Authors
I examined your work titled "Leveraging Vision Transformers for MRI-Guided Early Response of Brain Metastasis to Stage GKRS Treatment" in detail. In this study, the authors evaluated the performance of the Vision Transformer model in predicting early responses of brain metastases to radiosurgery, achieving an overall accuracy of 97%. Although the article is generally well organized, the literature review is not comprehensive enough. Although there are large overlaps in methodology, scope, and content with the authors' previously published works (doi:10.3390/diagnostics13172853), this work is not referenced at the points of overlap. Instead, a collective reference is given to their previous work [25] in the following sentence: "The findings align with previous studies highlighting the efficacy of deep learning models in radiological applications [22-26]." This situation creates a huge contradiction. The plagiarism rate should be reduced. The article as it is appears to be a slice of the existing publication. I would like to point out a few corrections.
1- The article contains many typos. Eg: "brain tissue. [3] However", "process [4].The". Errors should be corrected before resubmission.
2- There are inconsistencies between the line spacings of the texts in the article, these should be corrected. E.g: The last paragraph of the introduction, the first paragraph of section 2.4, etc.
3- There are many abbreviations that are not explained in the abstract. E.g.: SRS, BM, MRI, etc.. Abbreviations cannot be used before their explanations are given, (including in the title).
4- In order to understand the novelty and originality of the article more clearly, the superior aspects of the existing studies should be addressed more comprehensively. Contributions should be given as articles in the introduction section of the article.
5-There is no related studies section in the article. However, ViT applications in MRI image processing have been popular in recent years and there are many innovative studies. Among them, the advantages of the proposed approach and the shortcomings of the existing studies should be evaluated.
6-Some sections of the article are identical with the authors' previous studies. Ex: 2, 2.1, 2.2, 2.3 and 2.4 etc. What is the difference between this study and the previous study? The superior aspects that can be considered as innovations should be emphasized.
7-To ensure the generalizability of the model, the model needs to be validated using a larger dataset. Additionally, sensitivity analyses can be performed to see how data diversity affects model performance.
8- The study evaluates ViT performance. Comparisons of ML or different DL architectures can reveal in which aspects the ViT model is superior or weaker and support the selection of the model.
9- The distribution of misclassifications in the confusion matrix over patient groups can be examined in more detail. This can provide information about which patient types the model's classification performance is more reliable.
10- Not enough details were given about the dataset used for validation. For the generalizability of the results, it would be useful to work on more datasets or to add a discussion on whether the method can achieve similar performance on other datasets.
11- A quantitative evaluation is not made in the discussion section. It would be appropriate to emphasize the effectiveness of the method by evaluating the quantitative results in the discussion and results section.
Reviewer 2 Report
Comments and Suggestions for Authors
The authors have utilized a Vision Transformer (ViT) to classify MRI images into binary categories to predict early responses to SRS. However, the paper lacks methodological novelty, as it primarily applies an existing architecture (ViT) without introducing significant innovations or adaptations specific to this clinical context.
Figure 4 shows loss over the epoch. Which loss?
Mathematical framework for methodology and quantitative analysis is missing.
Few references are not available.